# Improved Production and Biophysical Analysis of Recombinant Silicatein-α

**DOI:** 10.3390/biom10091209

**Published:** 2020-08-20

**Authors:** Emily I. Sparkes, Rachel A. Kettles, Chisom S. Egedeuzu, Natalie L. Stephenson, Stephanie A. Caslin, S. Yasin Tabatabaei Dakhili, Lu Shin Wong

**Affiliations:** 1Manchester Institute of Biotechnology, University of Manchester, 131 Princess Street, Manchester M1 7DN, UK; emilysparkes11@gmail.com (E.I.S.); rachel.kettles@manchester.ac.uk (R.A.K.); chisom.egedeuzu@manchester.ac.uk (C.S.E.); natalie_stephenson@hotmail.co.uk (N.L.S.); stephanie.caslin@sky.com (S.A.C.); s.yasin.tabatabaei.d@gmail.com (S.Y.T.D.); 2Department of Chemistry, University of Manchester, Oxford Road, Manchester M13 9PL, UK

**Keywords:** silicatein, biocatalysis, biosilification, organosilicon, organosiloxane, hexahistidine tag, protein aggregation, protein engineering, protein production

## Abstract

Silicatein-α is a hydrolase found in siliceous sea sponges with a unique ability to condense and hydrolyse silicon–oxygen bonds. The enzyme is thus of interest from the perspective of its unusual enzymology, and for potential applications in the sustainable synthesis of siloxane-containing compounds. However, research into this enzyme has previously been hindered by the tendency of silicatein-α towards aggregation and insolubility. Herein, we report the development of an improved method for the production of a trigger factor-silicatein fusion protein by switching the previous hexahistidine tag for a Strep-II tag, resulting in 244-fold improvement in protein yield compared to previous methods. Light scattering and thermal denaturation analyses show that under the best storage conditions, although oligomerisation is never entirely abolished, these nanoscale aggregates of the Strep-tagged protein exhibit improved colloidal stability and solubility. Enzymatic assays show that the Strep-tagged protein retains catalytic competency, but exhibits lower activity compared to the His_6_-tagged protein. These results suggest that the hexahistidine tag is capable of non-specific catalysis through their imidazole side chains, highlighting the importance of careful consideration when selecting a purification tag. Overall, the Strep-tagged fusion protein reported here can be produced to a higher yield, exhibits greater stability, and allows the native catalytic properties of this protein to be assessed.

## 1. Introduction

The silicateins are a family of enzymes from marine demospongiae that are involved in biosilicification, the biogenic conversion of soluble silicates into inorganic silica [1,2]. Biochemically, the silicateins catalyse the hydrolysis and condensation of silicon-oxygen bonds, a capability unique to a small number of organisms [1,2]. Silicatein-α (Silα) is the most abundant of silicatein’s three isoforms that form axial protein filaments within the silica spicules of *Suberites domuncula* [3]. Analysis of the proteins’ primary sequence shows a high degree of similarity to proteases of the cathepsin family [3,4]. One major difference between these two families of enzyme, however, is found in the catalytic triad. Both proteins bear the classical Xaa-His-Asn motif, but while cathepsin L is a cysteine protease with this residue in the Xaa position, Silα bears serine instead. 

Silα’s ability to catalyse Si-O bond formation and hydrolysis make it an interesting candidate enzyme for applications in the synthetic manipulation of siloxane chemistry. This aspect is significant since siloxanes are used widely in modern life, occurring in lubricants, bulking agents, adhesives, home appliances and cosmetics [5,6,7]. They also play a major role as auxiliaries in the chemical synthesis of complex molecules [8,9,10]. Current methods of organosiloxane production, however, involve the use of environmentally undesirable chlorosilanes [11]. Using Silα to manipulate Si-O bonds while avoiding chlorinated feedstocks would potentially be much more environmentally sustainable, and could lead to the recycling and reuse of organosiloxanes.

One current limitation hindering the wider application of Silα is that it is highly prone to aggregation and poor solubility [12]. This issue has thus far presented a serious hindrance for the production, purification, and storage of the enzyme, hence its wider biotechnological application. It has been hypothesised that silicatein forms ordered oligomers to template the formation of biosilica spicules [13], which may contribute to its aggregative nature. Efforts have been made to develop a soluble form of Silα by fusing it to a variety of other proteins known to act as solubility aids, including most recently with Protein S (ProS2) [14,15]. However, these efforts were largely unsuccessful. In the case of the fusion with ProS2, soluble protein could be obtained, but only after a lengthy process of extraction from inclusion bodies and refolding. Thus far, the greatest success in the direct production of soluble Silα (i.e., without denaturation and refolding) has been achieved upon fusion to the chaperone protein trigger factor (TF), or through the use of buffers containing detergents such as Triton X-100 and 3-[(3-cholamidopropyl)dimethylammonio]-1-propanesulfonate (CHAPS) [15]. Even so, their solubility beyond a few hours remains marginal.

In these earlier studies, a hexahistidine (His_6_) tag was incorporated into the protein for the purposes of purification. However, this tag possesses properties that may also be contributing to the low production yield and stability. As this tag is used in conjunction with immobilised metal affinity chromatography (IMAC), organic buffers containing amines may compete for the metal. This issue limits the range of buffers, and hence pH range that can be employed [16]. Furthermore, it is known that His_6_ tags can negatively affect the solubility of proteins [17] and alter their physicochemical properties. Indeed, there have been reports that His_6_ tags can induce protein dimerization [18], introduce esterase activity [19], and reduce catalytic activity due to misfolding, steric hindrance, and electrostatic interactions [20,21,22,23].

This report details efforts to investigate the protein design, formulation, and purification procedures to increase overall protein production yield. These efforts involved comparing the effect of two different affinity tags, the His_6_ tag and Strep II tag (Strep) [16]. The tagged proteins which were isolated were subsequently subjected to further biophysical characterisation by circular dichroism (CD) spectroscopy, light scattering, and thermal denaturation, as well as biochemical assays of enzyme activity.

## 2. Materials and Methods 

All solvents and reagents were of analytical grade and purchased from either Sigma-Aldrich, VWR or Fisher Scientific. The 4-*tert*-butyldimethylsilyloxynitrobenzene (TBDMS-ONp) substrate was chemically synthesised from the corresponding silyl chloride and 4-nitrophenol as previously reported [15]. All polymerases, ligases and restriction enzymes were obtained from New England Biolabs, as were the DNA ladders (1 kb) and 6x DNA loading dye. Nucleic acid isolations were conducted using the PCR Purification Kits and Miniprep plasmid extraction Kits supplied by Qiagen (Hilden, Germany). Codon-optimised DNA encoding Silα and TF-Silα-Strep were produced by GenScript (Piscataway, NJ, USA) and Integrated DNA Technologies (San Diego, CA, USA), respectively. The *Escherichia coli* (*E. coli*) BL21(DE3) cells were supplied by Novagen (Darmstadt, Germany).

Cell lysis was conducted using a Bandelin Sonoplus HD2070 probe sonicator. IMAC was carried out using gravity flow columns (Qiagen, Hilden, Germany) loaded with Ni-NTA agarose resin (Thermo Scientific, Waltham, MA, USA). Streptavidin affinity chromatography (SAC) was carried out using StrepTrap HP columns (GE Healthcare, Chicago, IL, USA) on an ÄKTApurifier system (GE Healthcare).

CD measurements were performed on a Chirascan CD spectrometer (Applied Photophysics, Leatherhead, UK) in quartz cells with a 0.1 mm pathlength (Starna Scientific, Ilford, UK). Dynamic light scattering (DLS), static light scattering (SLS) and intrinsic fluorescence data were measured simultaneously on an UNcle instrument (Unchained Labs, Pleasanton, CA, USA). 

### 2.1. Gene Cloning and Protein Overexpression

The gene cloning and protein overexpression of His_6_-Silα and His_6_-TF-Silα proteins (see below for explanation of protein design) were carried out as previously reported [15]. For the isolation of overexpressed protein, the frozen cell pellet was resuspended in the appropriate lysis buffer (Table 1). The resuspended cells were lysed by sonication at 4 °C for 8 cycles of 1 min pulse and 5 s rest at 50% amplitude. The lysate was centrifuged (48,200× *g* for 30 min at 4 °C) and the supernatant collected. The soluble fractions and pellet were analysed by SDS-PAGE to determine the location of the protein of interest. Where the protein of interest was in the soluble fraction, it was isolated from the supernatant by IMAC, eluting with increasing imidazole concentrations (20–250 mM) in the same buffer used for lysis through a gravity flow column containing 5 mL of gel.

For TF-Silα-Strep, a chemically synthesised codon-optimised DNA fragment encoding the fusion protein with a 5′ NdeI site and 3′ BamHI site was used as the starting point. The tf-silα-strep gene was digested with NdeI and BamHI and ligated into a pET11a vector. Successful molecular cloning of the completed pET11a-TF-Silα-Strep was confirmed by DNA sequencing. BL21(DE3) cells transformed with this vector were grown at 37 °C overnight, in LB medium with 100 µg/mL ampicillin. This resulting culture (10 mL) was used to inoculate 800 mL fresh LB medium, which in turn was grown at 37 °C to an optical density (OD600) of 0.6. IPTG was added to a final concentration of 1 mM and the culture shaken overnight at 16 °C. Cells were harvested by centrifugation (3500× *g* for 20 min at 4 °C), the media discarded, and the cell pellet frozen at −20 °C prior to lysis. The frozen cell pellet was resuspended in the appropriate lysis buffer (Table 2). The resuspended cells were lysed by sonication at 4 °C for 8 cycles of 1 min pulse and 5 s rest at 50% amplitude. The lysate was centrifuged and analysed in the same manner as the His_6_-tagged proteins. The TF-Silα-Strep was isolated from the supernatant using 5 mL StrepTrap columns (with multiple columns connected in series where necessary for larger batches), with elution of the protein from the column being performed using the same buffer used for lysis with an additional 1 mM EDTA and 2.5 mM desthiobiotin.

In all cases, the concentration of the isolated proteins were estimated by UV-vis absorbance at 280 nm using molar absorption coefficients computed by ProtParam [24,25], and the yield was quantified by multiplying the volume of solution and the concentration of the protein.

### 2.2. Circular Dichroism

All protein samples were exchanged into the relevant buffer (Table 3) by gel filtration using a PD-10 column, then diluted to 1 mg mL^−1^ for analysis. Data was collected over a wavelength range of 192 to 260 nm, at a scan speed of 0.5 nm s^−1^. The temperature was maintained at 22 °C using a temperature-controlled chamber purged with N_2_. Baseline measurements were performed using buffer alone. The raw CD data was used to calculate molar ellipticity and mean residue molar ellipticity.

### 2.3. Dynamic Light Scattering, Static Light Scattering and Intrinsic Fluorescence

All protein samples were exchanged into the relevant buffer by gel filtration using a PD-10 column, then diluted to 1 mg mL^−1^ with relevant buffer (Table 3). 8 μL of each sample was loaded into the multi-micro cell array in triplicate. The data from individual experiments were inspected, any outliers discarded, and the remaining data averaged. All three samples were analysed within 2 h of buffer exchange. The temperature was increased from 18 to 90 °C at a rate of 2 °C min^−1^. The SLS and fluorescence measurements were recorded every 30 s, while the DLS measurement was recorded at 20 °C.

DLS measurements (size distribution and polydispersity) were calculated with the UNcle software (version 2.0) correlation function. The hydrodynamic radius (*d_h_*) amplitude maxima were then identified in R using the stat_peaks function from the ggpmisc package [26], and the *d_h_* for each protein calculated as the mean *d_h_* at maximum amplitude ± standard deviation across the three pH conditions. The estimated theoretical hydrodynamic diameters of folded proteins were calculated using the following equation [27]:(1)$$d_{h}=\frac{2\left(4.75\;\pm \;1.11\right)N^{0.29\;\pm \;0.02}}{10}$$
where *d_h_* is the hydrodynamic diameter in nm, and *N* is the number of residues in the polypeptide chain. For expected hydrodynamic diameters of unfolded proteins, the following equation was used [27]:(2)$$d_{h}=\frac{2\left(2.21\;\pm \;1.07\right)N^{0.57\;\pm \;0.02}}{10}$$

For SLS, the intensity of scattered light was measured at 266 nm and was used to calculate the aggregation temperature (*T_agg_*) with UNcle software (version 2.0). The intrinsic fluorescence emission was measured at 473 nm and the first derivative of the barycentric mean (BCM) was used to calculate the melting temperature (*T_m_*) using the same software. 

### 2.4. Enzyme Activity Assays

Tetraethoxysilane (TEOS) assays were undertaken following the procedure published previously [15]. The TBDMS-ONp assays were based on that which was previously published, but modified to compensate for the lower activity of the Strep-tagged protein. 

In this case, a 1 mM stock of TBDMS-ONp was prepared in aqueous solution with 10% *v/v* 1,4-dioxane. A series of 2 mL working solutions at 10-fold the concentration of substrate required for each enzyme assay was then made by diluting the stock solution with the appropriate volume of 10% *v*/*v* aqueous 1,4-dioxane. 20 μL of each working solution was aliquoted into each well of a microtitre plate and 80 μL of the assay buffer (50 mM Tris, 100 mM NaCl at pH 8.5) was added. Separately, the enzyme was buffer exchanged into the same assay buffer from the purification buffer by overnight dialysis and adjusted to a concentration of 13.42 μM. 100 μL this enzyme solution was added and mixed to initiate the reaction (i.e., final assay volume of 200 μL and final enzyme concentration of 6.71 μM). The spectrometric data collection was then commenced and the UV-Vis absorbance at 405 nm was recorded every 5 min at ambient temperature (approximately 22 °C) for 1200 min with periodic shaking. Each time-course measurement was carried out in triplicate.

The concentration of the nitrophenoxide ion released during the reaction was determined from a calibration curve of known concentrations of 4-nitrophenol in the same reaction buffer. Control reactions were carried out where the enzyme solution was omitted and replaced with an equivalent volume of the assay buffer (50 mM Tris, 100 mM NaCl at pH 8.5). The concentration of the phenoxide product from the control reactions were then subtracted from each enzymatic reaction, to obtain the net product concentration generated by enzyme catalysis. Graphs of net phenoxide concentration against time were then plotted. The initial velocities (*V*_0_) were obtained from performing a linear regression of the initial part of the curve (typically the first 50 min of the assay, which is assumed to approximate a linear increase). The *V*_0_/[E] was then plotted against the corresponding substrate concentration in each assay reaction and fitted against the equation for the Michaelis-Menten model of enzyme kinetics within the OriginPro 2019b (build 9.6.5.169) software to obtain the *K_M_* and *V_max_* values. The *k_cat_* values were calculated from the *V_max_* and the enzyme concentration, using the formula *k_cat_* = *V_max_*/[E].

## 3. Results and Discussion

### 3.1. Design of Recombinant Protein Constructs

Three separate protein constructs were designed and the genes synthesised for: (i) mature (without its pro-peptide) Silα from *S. domuncula* fused to an *N*-terminal hexahistidine tag (His_6_-Silα); (ii) mature Silα fused at the *N*-terminal to TF, itself fused at the *N*-terminal to an His_6_ tag (His_6_-TF-Silα); and (iii) mature wild type Silα fused at the *N*-terminal to TF and a *C*-terminal Strep II tag (TF-Silα-Strep) (Figure 1). The two His_6_-tagged proteins were the same as those in the previous report [15], with the TF-fusion already having been shown to exhibit superior performance in terms of practical handling and stability. The Strep II tag was chosen for investigation as it was known to be unaffected by the choice of buffer (as long as the pH remains above 7), and the presence of a wide range of additives such as metal ions and chelating agents [16,28,29]. In all cases, gene fusions were cloned into a synthetic vector under the control of the T7*lac* promoter, and transformed into *E. coli* BL21(DE3). The proteins of interest were then overproduced according to standard procedures.

### 3.2. Optimisation of Silα Purification

The His_6_-Silα was first investigated, as this protein most closely resembled the wild type enzyme, but it had the poorest solubility [15]. Here, its solubility during cell lysis and subsequent chromatographic isolation was assessed with a range of buffer formulations (Table 1). Any buffer formulation(s) that gave appreciable amounts of soluble protein upon lysis (as evidenced by SDS-PAGE analysis of the soluble lysate fraction) were then used for the subsequent affinity chromatography to isolate the protein.

In general, these buffer formulations were chosen to investigate the effect on protein solubility of small molecule additives and pH. The addition of non-denaturing detergents such as Triton-X and CHAPS was investigated as these have been previously used in earlier reports on the isolation of His_6_-Silα [15,30]. The addition of amino acids L-arginine and L-glutamic acid was also investigated as they have been shown to reduce protein aggregation, increase thermal stability and solubility; through increasing the surface tension of water to allow preferential hydration of proteins and by acting as weak surfactants [31,32,33]. The pH range investigated was based on findings that a higher pH may disrupt hydrophobic interactions between the protein units [13].

For His_6_-Silα it was found that the majority of the buffers tested did give soluble protein (Table 1, entries 4–6 and 8–13; Appendix A), but only the buffer containing Tris and NaCl at pH 8.5 gave soluble purified protein fractions after IMAC (Table 1, entry 8). This result is consistent with previous reports that only the weakly coordinating buffer Tris would be compatible with IMAC [16]. In comparison, earlier work investigating the same buffer but at pH 7.5 gave no soluble protein [15] (Table 1, entry 7). The addition of Arg and Glu appeared to give no benefit in this case. These results show that pH had a greater effect on protein solubility than the presence of detergents or buffer additives. Even so, the isolated yield was poor with only approximately 0.2 mg (equivalent to 8 nmol) of purified His_6_-Silα obtained per litre of *E. coli* culture (Table 4). 

For His_6_-TF-Silα, the best performing buffer identified above (Table 1, entry 8) was then compared to the conditions used in previous studies [13], where the use of 100 mM phosphate buffer at pH 8.0 during lysis and isolation gave only approximately 0.5 mg per litre of culture. In contrast, the new formulation exhibited higher levels of protein in the soluble fraction (Figure 2a) and also gave higher isolated yields. At pH 8.5, 14 mg (177 nmol) per litre of culture was achievable, which was a 28-fold molar improvement compared to the phosphate buffer at pH 7.0, and 22-fold superior to His_6_-Silα noted above.

However, despite this increased production yield, aggregates were regularly observed for both His_6_-tagged proteins upon visual inspection in the purified fractions immediately after isolation by IMAC, suggesting the solubility was still relatively poor. The estimated pI of His_6_-Silα and His_6_-TF-Silα and are respectively 5.65 and 5.13 (calculated using Compute pI/Mw [25,34]), so raising the pH of the buffer is expected to increase the electrostatic repulsion in both proteins and allow for better solubilization [35,36]. However, higher pH levels are incompatible with IMAC.

The TF-Silα-Strep construct was then evaluated as it had a calculated pI of 4.92, and thus was hypothesised to have an increased solubility at higher pH. In addition, SAC tolerates a greater range of buffer conditions compared to IMAC, making purification in increasingly basic buffers possible. Thus, an analogous buffer survey was performed for this protein (Table 2). Here, it was found that all the AMP-containing buffers at higher pH (Table 2, entries 5–9), previously incompatible with IMAC, now facilitated soluble protein production (Figure 2b). Subsequent purification by SAC yielded 115 mg (1.54 μmol) per litre of culture with the best AMP-containing formulation at pH 9.0 (Table 2, entry 6), representing a 244-fold molar improvement compared to His_6_-TF-Silα when the phosphate buffer was used previously at pH 7.0, and 8.7-fold improvement compared to the best His_6_-TF-Silα result shown above (Table 4). The buffer consisting of Tris and NaCl at pH 8.5 also gave soluble protein (Table 2, entry 4; Figure 2b). In this case, a somewhat lower yield of 58 mg (777 nmol) per litre of culture was obtained. Nevertheless, in all cases when this Strep-tagged protein was purified at pH ≥ 8.5, no visible aggregates were ever observed in protein-containing fractions after chromatography and concentration. 

These results therefore substantiate the hypothesis that greater solubility of the Silα constructs are achieved when the pH is significantly raised above the pI. These results may also indicate that the presence of the His_6_-tag negatively affects their solubility. If so, this result would be consistent with reports on other types of proteins bearing this tag [17,18,23]. Indeed, the His_6_-Silα exhibited extremely poor solubility and almost entirely precipitated from solution within hours, and was not further investigated.

### 3.3. Circular Dichroism Spectroscopy

To ascertain if the proteins were correctly folded after isolation, the CD spectra of both His_6_-TF-Silα and TF-Silα-Strep proteins were recorded. These analyses were performed at pH 7.0, 8.5, and 9.0 (Table 3). For pH 7.0, a phosphate buffer containing KCl was used as it has previously been used for lyophilisation of the protein [15]. The other two buffers were chosen from the best results in the protein production experiments above.

In all cases, the proteins showed clear secondary structural features (Figure 3). Strong signals corresponding to alpha helices (negative at 222 and 208 nm, and positive at 193 nm) were present, suggesting that the two protein constructs are mainly alpha-helical. All spectra have line shapes that are essentially identical to those previously recorded for His_6_-TF-Silα [15], apart from the large signal at 192 nm for the spectra measured at pH 9.0, which is due to the presence of the AMP buffer. The data indicates the proteins are not denatured or disordered under all the tested conditions [37].

### 3.4. Dynamic Light Scattering

His_6_-TF-Silα and TF-Silα-Strep were then analysed by light scattering to provide a quantitative analysis of protein aggregation. These analyses were performed using the same conditions as for the CD spectroscopy (Table 3). 

Firstly, dynamic light scattering (DLS) analyses were performed to estimate the particle size distribution (expressed as hydrodynamic diameter, *d_h_*; Figure 4). For His_6_-TF-Silα, highly heterogeneous distributions were observed under all the tested conditions, whilst for the Strep-tagged protein only a single peak is seen in all cases. The major peaks for each proteins sit at 89.4 ± (a standard deviation of) 5.5 nm for His_6_-TF-Silα and 29.5 ± 3.6 nm for TF-Silα-Strep, when averaged across the three conditions. TF and Silα are individually estimated to have *d_h_* of less than 10 and 4.6 nm, respectively [13,38]. Using empirical formulae relating polypeptide length and *d_h_* proposed by Wilkins et al. [27], the predicted *d_h_* of His_6_-TF-Silα and TF-Silα-Strep are respectively 6.5 nm and 6.4 nm when folded (assuming a globular structure); and 19.0 nm and 18.4 nm if unfolded. These calculated values are much smaller than those recorded by DLS, even when considering the fact that *d_h_* values from DLS are based on an assumption of globular structure and thus cannot provide a completely reliable reflection on the size of non-globular proteins such as TF [39].

These findings can thus only be explained by the formation of oligomeric structures by both proteins, and is consistent with the well-known propensity for Silα to self-assemble [13]. The DLS data also suggests oligomer formation is more pronounced for His_6_-TF-Silα, with the presence of wide size distributions or multiple peaks suggesting a broad mix of oligomers [40].

Using the DLS data, the percentage polydispersity and polydispersity index (PDI) were calculated for each sample to quantify their homogeneity (Table 5) [41,42]. In comparing the two proteins, His_6_-TF-Silα gives consistently higher values across all tested conditions. For both proteins, results at pH 9.0 show the lowest percentage polydispersity and PDI, indicating lower aggregation and supporting the hypothesis of electrostatic repulsion between protein molecules. However, the percentage polydispersity for both proteins at pH 8.5 was higher compared to pH 7.0, which was unexpected since the SDS-PAGE analysis post-lysis suggested good solubility at the higher pH (Figure 2). One possible explanation for the lower observed polydispersity at pH 7.0 could be that aggregation and subsequent precipitation of the protein from solution had occurred prior to DLS analysis. As the light scattering measurements do not detect precipitated material, the observed (apparently lower) polydispersity would be based only on the residual protein that remained in solution. 

The PDI of TF-Silα-Strep decreases with increasing pH, and is always below 0.35, indicating a high degree of homogeneity. Thus, whilst the percentage polydispersity of this protein suggests aggregation is occurring, the low PDI suggests that these aggregates are relatively uniform in size. In contrast, the PDI of His_6_-TF-Silα is never below 1. It is evident, therefore, that the His_6_-tagged variant is prone to form a wider range of aggregates under all the tested buffer conditions. 

### 3.5. Temperature-Dependent Melting 

To further complement the DLS data, the thermal stability of both proteins was assessed by determining their melting temperature (*T_m_*). Both protein constructs were subjected to a thermal ramp in the aforementioned buffer conditions (Table 3), and protein denaturation was quantified by the change in peak wavelength of the intrinsic tryptophan fluorescence emission (expressed as change in the barycentric mean wavelength, BCM). Here, denaturation results in the exposure of the internal tryptophan residues to the more polar aqueous media and a corresponding bathochromic shift in emission wavelength. The *T_m_* is then determined from the first derivative of the BCM as a function of temperature. 

For His_6_-TF-Silα (Figure 5a), measurements at all three conditions showed a gradual increase in BCM with temperature without any sharp transitions, rather than a classical sigmoidal line shape with a single inflection corresponding to the *T_m_*. Nevertheless, an apparent *T_m_* can be estimated based on the maxima in the first derivative (Figure 6). As the CD spectra displayed the expected features, the lack of sharp transitions is unlikely to be due to the protein already being unfolded or misfolded. A more likely explanation is that the protein sample consists of a heterogeneous population of aggregates (as evidenced by the DLS data), and the plot represents the ensemble average with no single *T_m_*.

In contrast, the data for TF-Silα-Strep show BCM step transitions giving apparent *T_m_* values of approximately 51.8, 49.0 and 45.8 °C at pH 7.0, 8.5 and 9.0, respectively. These transitions are overlaid with a gradual increase in BCM throughout the entire temperature range that is possibly due to subpopulations of aggregated protein dissociating with the rising temperature (Figure 5b). 

### 3.6. Temperature-Dependent Aggregation

To deconvolute the two phenomena of protein unfolding and aggregation, static light scattering (SLS, at 266 nm) measurements were performed as a function of increasing temperature, as it allows quantification of the average molar mass of the particles. Here, the SLS intensity is proportional to the molar mass of the particles in solution, and can thus be used to infer aggregation. In this analysis, SLS signal as a function of temperature are plotted and the aggregation temperature (*T_agg_*) can be assigned from the first point that exceeds two standard deviations above the baseline signal at the start of the experiment (at 20 °C). 

For His_6_-TF-Silα (Figure 5c) the data approximates a broad sigmoidal shape at pH 8.5 and 9.0, with apparent *T_agg_* at 45.6 and 45.9 °C, respectively. The transition appeared to be sharper at pH 9.0, suggesting a more homogeneous population that aggregates over a narrower temperature range. Furthermore, the signal also plateaus at a higher level, indicating the formation of larger aggregates compared to pH 8.5. In the case of pH 7, above approximately 60 °C there is a sharp rise followed by fluctuating signal intensity, which suggests a large increase in particle size leading to macroscopic precipitation. 

In the case of TF-Silα-Strep (Figure 5d), it can be seen that the signals at both pH 8.5 and pH 7.0 start at higher signal intensity than pH 9.0, suggesting protein particle sizes are already larger for those cases at the start of the experiment. This result is consistent with the DLS data (Figure 4b) that shows a wider size distribution at pH 8.5 and 7.0. Only the data at pH 7.0 appeared to give a sigmoidal shape, with a *T_agg_* at 52 °C. At pH 8.5, there is a gradual increase in SLS signal, overlaid with a very weak transition at an apparent *T_agg_* of 45 °C. At pH 9.0, the data exhibited complex behaviour, with a gradual decrease until about 40 °C, an apparent transition at approximately 45 °C, followed by a gradual rise thereafter. The decrease in signal could be due to heat-induced increases in solubility and disaggregation. As the temperature increases towards 90 °C, at pH 9.0 this protein appears to form larger aggregates than those at pH 8.5, though the high signal intensity suggests that these particles are still stably suspended in solution. The proteins in pH 7.0 display the highest signal intensity at 90 °C and thus are forming the largest aggregates, though they apparently continue to remain suspended in the buffer solution under these conditions. 

In comparing the *T_m_* and *T_agg_*, for His_6_-TF-Silα (Figure 6a) no clear correlations or trends were observed, as the data generally showed a lack of clear transitions or macroscopic precipitation that confounded further analysis (Figure 5). In contrast, TF-Silα-Strep exhibited good *T_m_* and *T_agg_* agreement at all the tested pH values (Figure 6b). This correlation shows that heat-induced unfolding and aggregation occur concurrently, and is consistent with the general theory that protein aggregation is caused by denatured or misfolded protein. Furthermore, *T_m_* and *T_agg_* decrease with increasing pH for TF-Silα-Strep, suggesting that the protein is more stable towards thermally induced denaturation at neutral pH. However, the DLS data show that the protein is more aggregated at neutral pH, as evidenced by a wider size distribution compared to pH 9.0 (Figure 4b). Taken together, these results suggest that, at pH 7.0 under ambient temperatures, the protein already presents as oligomers, which then undergo denaturation and further aggregation upon thermal treatment. Thus, in this case a basal level of aggregation already manifests under ambient conditions that is not due to the presence of denatured protein. This behaviour is consistent with the fact that Silα has many hydrophobic residues already present on the exterior [13], which will promote protein-protein interactions even with correctly folded proteins.

In comparing His_6_-TF-Silα and TF-Silα-Strep, the presence of the His_6_ tag appears to have a further detrimental effect on solubility. This observation is consistent with other reports suggesting its presence negatively influences solubility through a variety of effects including increased hydrophobicity, the formation of dimers or promotion of protein misfolding [18,20,22,43]. Indeed, the His_6_-tagged protein is more predisposed to aggregation and forms macroscopic filament-like structures (that can even be seen by the naked eye) within minutes of isolation, even at pH 9.0. Nevertheless, the same general findings can be inferred, in that the His_6_-tagged protein presents as oligomers under ambient conditions, which undergo unfolding and further aggregation upon heating. 

Previous studies by Murr and Morse [13] suggested that aggregation of wild type Silα (without trigger factor) is a twofold process. The protein molecules first associate by hydrophobic interactions to form oligomers, which are subsequently crosslinked by intermolecular disulfide bonds. These oligomers in turn associate with each other through further hydrophobic interactions to form higher order structures, which can be disrupted upon exposure to high pH (pH ≥ 9.0). If so, the use of high pH buffers for protein purification and storage would only prevent these higher order species but would not yield monomeric proteins. 

Another potential contributor to aggregation could be the presence of C5 hydrogen bonding, which occurs through an overlap of the carbonyl *n_p_* and amide σ* orbitals between small amino acid residues such as glycine and serine. The computational models previously reported [12,13] indicate the presence of a high number of both serine and glycine residues on the solvent-accessible surface of Silα, which may result in a significant contribution of this type of bonding towards protein-protein interactions.

Nevertheless, these results indicate that the optimal buffer conditions for thermal stability and minimisation of protein aggregation are Tris (50 mM) and NaCl (100 mM) at pH 8.5, which provides a robust starting point for further applications.

### 3.7. Enzymatic Activity

To complete the characterisation of the proteins, their catalytic activity against Si-O bond hydrolysis were compared. Two hydrolytic activity assays were assessed, with TEOS and TBDMS-ONp [15,44].

The TEOS assays showed both candidates catalysed the production of silica compared to the negative control (buffer only), but the activity of TF-Silα-Strep was approximately 6.6-fold lower than that of His_6_-TF-Silα (Figure 7). These assays suggest (but do not confirm) that the His_6_ tag may also be enhancing catalytic activity of this protein towards silyl hydrolysis, and is consistent with reports that the tag can act as a catalyst for ester hydrolysis [19], and that polyhistidine groups can catalyse Si-O bond hydrolysis [45,46]. In contrast, previous studies with a range of proteins that do not contain a His_6_-tag did not display any catalysis towards the hydrolysis of Si-O bonds [15,47]. 

Since it is not expected that the His_6_-tag would be dependent on the tertiary structure of the protein for its catalytic activity, the relative contribution of protein folding (and thus the correctly constituted active site) was investigated by repeating the assays using equivalent amounts of heat-denatured proteins. In both cases, heating at 85 °C for 20 min prior to the assay results in a loss of 85 and 76 % of their activity compared to enzymes that were not heat-treated, for His_6_-TF-Silα and TF-Silα-Strep respectively. Thus, assuming the His_6_-tag is not buried within the bulk protein (either due to misfolding or aggregation) upon cooling prior to the assay, this result suggests that at least 10% of His_6_-TF-Silα’s activity is due to the His_6_-tag. In comparison, the Strep-tagged protein displays only a trace level of activity after denaturation.

Using TBDMS-ONp as the substrate, colorimetric assays were carried out and the Michaelis-Menten kinetic parameters were determined for the Strep-tagged protein and compared with the His_6_-tagged protein (Table 6, Appendix A). In comparing the two proteins, Strep-tagged protein displays a weaker affinity to the substrate, as evidenced by the higher *K_M_*. This result is consistent with the proposal that the His_6_-tag contributes to catalysis and would present a second binding site for the substrate, resulting in an apparently lower *K_M_*. Accordingly, the *k_cat_* of the Strep-tagged protein is also significantly lower as the catalysis is now only attributable to the enzyme active site.

Taken together, both assays suggest that the presence of the His_6_-tag on Silα contributes to non-active site mediated activity. Since the Strep-tag is not known to cause an equivalent effect, it is proposed that the fusion protein incorporating this tag gives a more accurate appraisal of catalytic activity attributable to enzyme itself, rather than the pendant tag. A possible alternative explanation is that the Strep-tag may be inhibiting catalytic activity. However, as the *C*-terminal location of this tag is distant from the active site, any possible inhibition would not be through direct steric hinderance.

## 4. Conclusions

In summary, a series of biophysical investigations were carried out with the aim of developing improved variants and formulations of the silicatein enzyme, which can serve as a starting point for future enzymology studies and biotechnological applications. During this study, the solubility and stability of three different protein constructs (His_6_-Silα, His_6_-TF-Silα, and TF- Silα-Strep) in various buffers were investigated. As expected, the His_6_-Silα without the TF fusion to aid solubility exhibited a high degree of aggregation and insolubility. For the remaining two candidates, increasing the pH of the buffers during lysis and purification resulted in a dramatic increase in soluble protein yield. Subsequent light scattering and temperature ramp analyses in the optimised buffers found that the His_6_-TF-Silα exhibited a lower solubility and colloidal stability compared to the Strep-tagged variant. Though not conclusive, these results together with others mentioned above present an emerging picture whereby the presence of a His_6_ tag may be detrimental to the solubility of some proteins. However, despite being more macroscopically soluble (and less prone to precipitation), TF-Silα-Strep was still found to be forming oligomeric structures even under the best conditions reported here. 

Assays of enzyme activity for both the TF-fused proteins show that the presence of a His_6_ tag may be contributing to a portion of the observed catalysis, though the TF-Silα-Strep protein still demonstrated unambiguous catalysis with respect to Si-O bond hydrolysis. This result illustrates that caution should be exercised when selecting purification tags, so that they do not unintentionally alter the activity of the recombinant protein. 

Further work will be needed to engineer the interactions contributing to the oligomerisation of Silα, which other reports have suggested include both hydrophobic interactions and intermolecular disulfide bonds. Thus, the generation of monomeric formulations of Silα could be achieved by the addition of additives that disrupt protein-protein interactions [48]. The intermolecular disulfide linkages can also be cleaved by the addition of reducing agents, though caution must be exercised as previous sequence analyses have indicated that internal disulfide bonds are also present that may be essential to the protein’s structure. Further experiments may also be carried out with an untagged protein, to exclude the possibility that the Strep-tag may be acting as an inhibitor to catalysis.

## Figures and Tables

**Figure 1 biomolecules-10-01209-f001:**
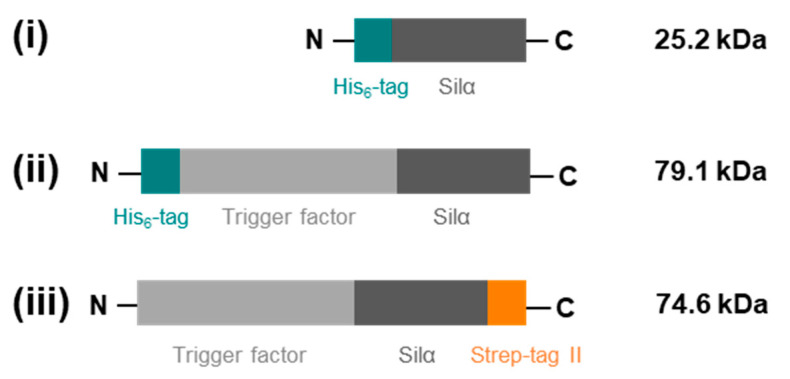
Schematic diagram showing protein constructs created: (**i**) His_6_-Silα; (**ii**) His_6_-TF-Silα; (**iii**) TF-Silα-Strep. For convenience, N- and C-termini are indicated, together with the molecular weight of each fusion protein.

**Figure 2 biomolecules-10-01209-f002:**
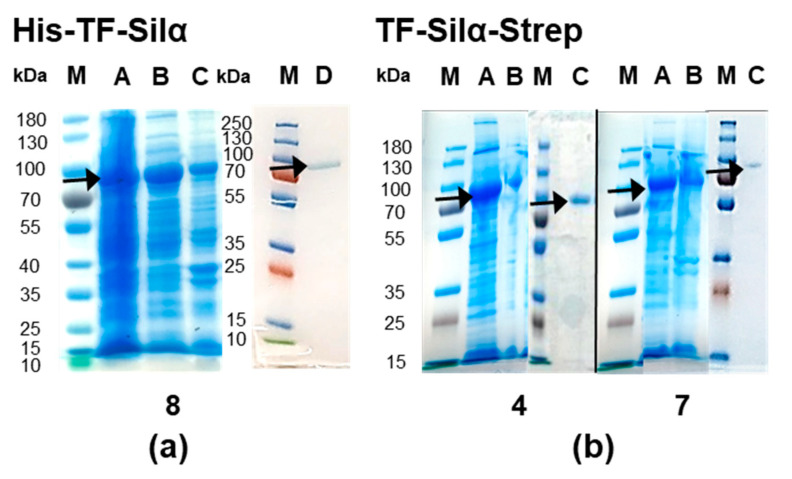
Images of SDS-PAGE gels demonstrating solubility analysis for (**a**) His_6_-TF-Silα; and (**b**) TF-Silα-Strep. For (**a**), the buffer stated in entry 8 of Table 1 was used; for (**b**), the buffer mixtures stated in Table 2 were used (numbers below each image correspond to the buffer entries in Table 2). Lanes for (**a**) are labelled as follows: M = marker, A = total lysed protein, B = soluble lysis fraction, C = insoluble lysis fraction, D = post-IMAC fraction. Lanes for (**b**) are labelled as: M = marker, A = soluble lysis fraction, B = insoluble lysis fraction, C = post-SAC fraction. Arrows are shown to indicate the presence of the desired protein band.

**Figure 3 biomolecules-10-01209-f003:**
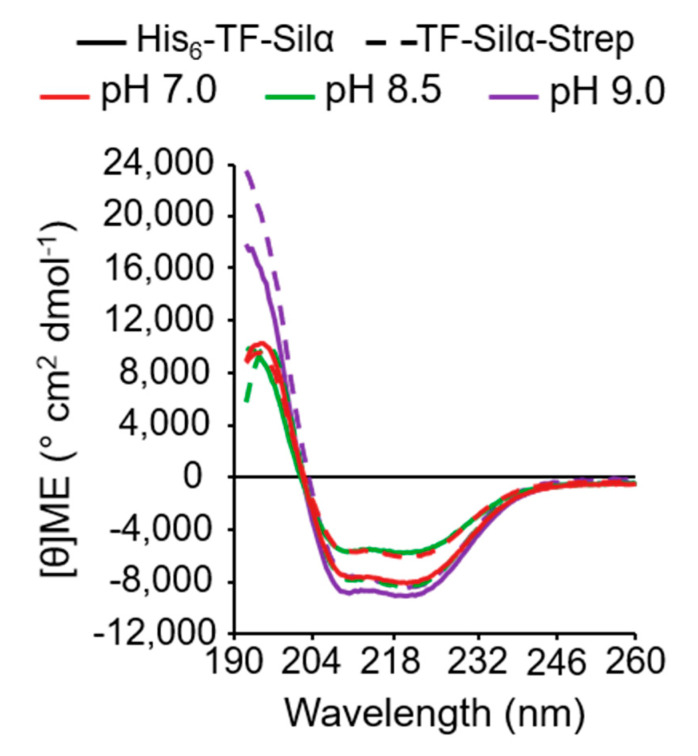
Circular dichroism spectra for His_6_-TF-Sila (solid line) and TF-Silα-Strep (dashed line) at pH 7.0, 8.5 and 9.0, using the buffers noted in Table 3.

**Figure 4 biomolecules-10-01209-f004:**
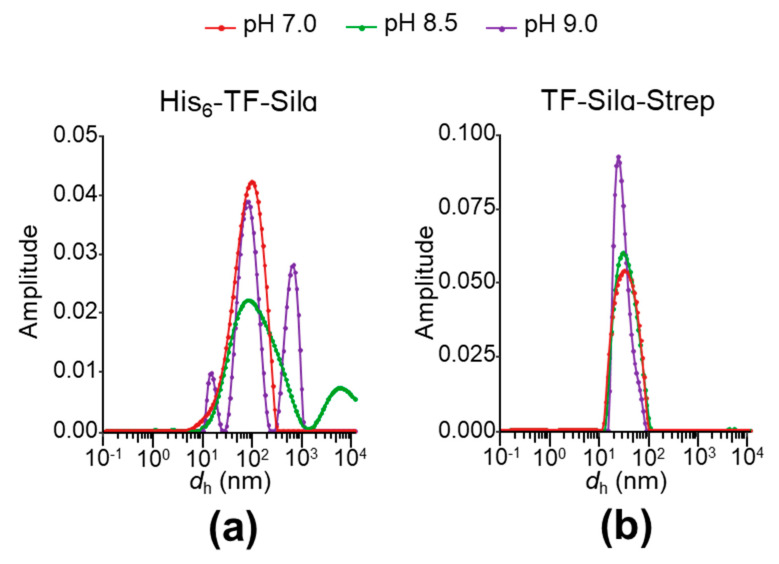
Dynamic light scattering measurements of (**a**) His_6_-TF-Silα; (**b**) TF-Silα-Strep. Graphs of amplitude against hydrodynamic diameter (*d**_h_*) for the respective proteins in the buffers listed in Table 3 are shown.

**Figure 5 biomolecules-10-01209-f005:**
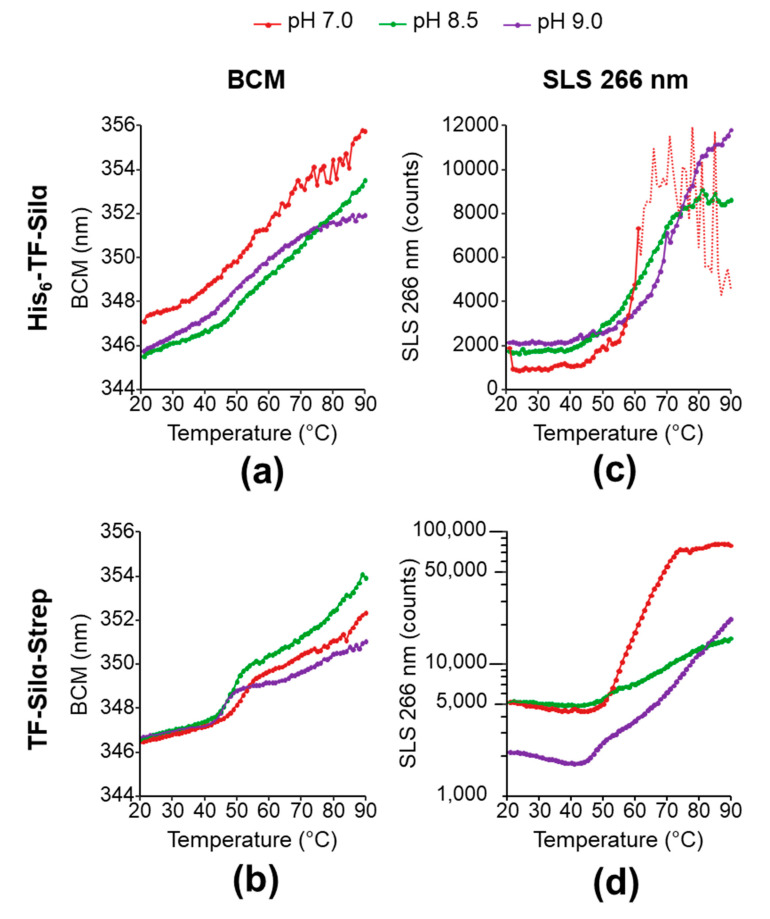
Data from a thermal ramp of silicatein protein constructs: (**a**,**b**) show the barycentric mean (BCM) of the intrinsic fluorescence spectra for His_6_-TF-Silα and TF-Silα-Strep respectively; (**c**,**d**) show the static light scattering (SLS) at 266 nm during the thermal ramp for His_6_-TF-Silα and TF-Silα-Strep respectively. The dotted line in (**c**) indicates measurements that are likely to be erroneous due to precipitation (see discussion for details).

**Figure 6 biomolecules-10-01209-f006:**
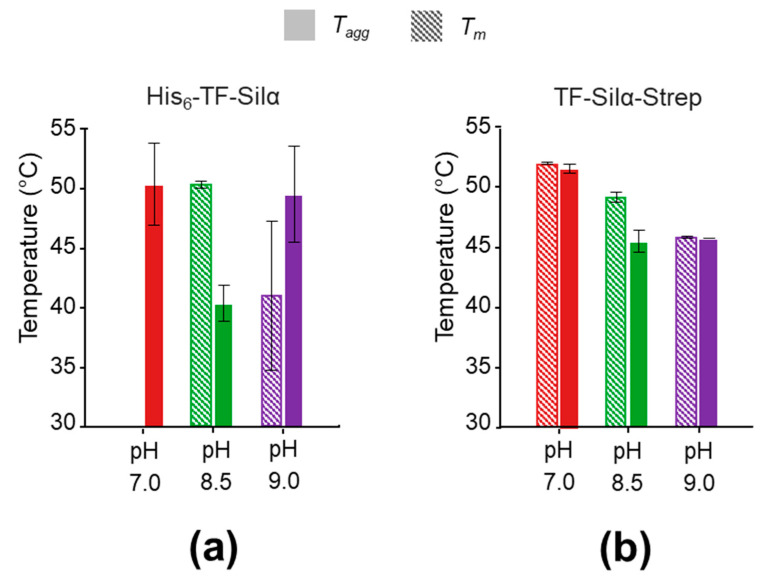
Graphs showing the apparent *T_m_* (hashed bars) and *T_agg_* (solid bars) calculated from SLS and fluorescence data for (**a**) His_6_-TF-Silα; (**b**) TF-Silα-Strep. Error bars shown are based on the standard deviation of two replicate protein samples.

**Figure 7 biomolecules-10-01209-f007:**
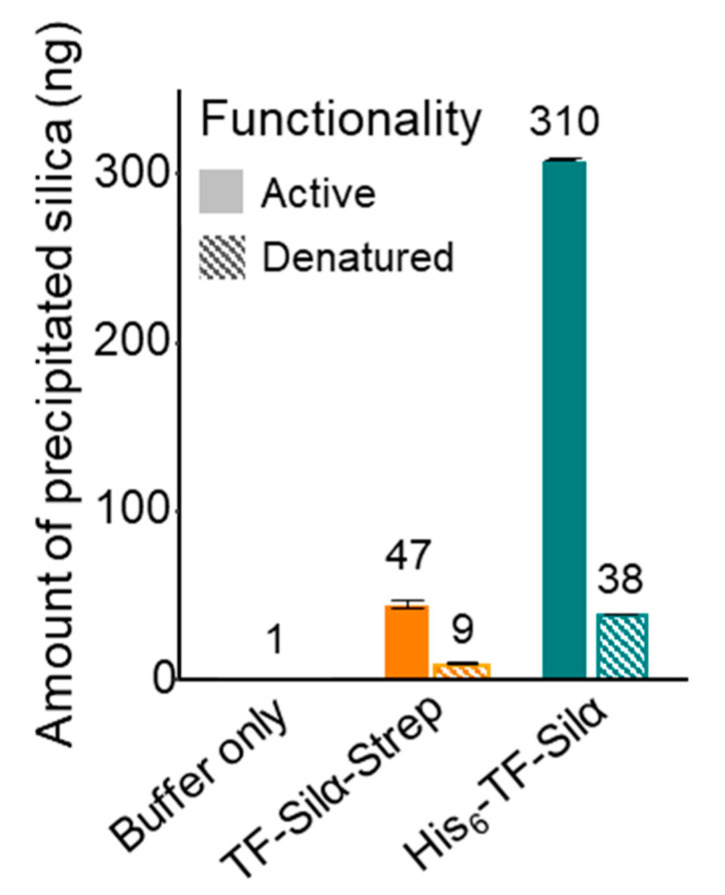
Graph showing amount of silica produced after 1 h, from the hydrolysis of TEOS by fresh (solid bars) and heat denatured (hatched bars) His_6_-TF-Silα and TF-Silα-Strep. Error bars are calculated based on the standard deviation of triplicate experiments.

**Table 1 biomolecules-10-01209-t001:** Buffer compositions investigated during cell lysis and IMAC purification of His_6_-Silα.

Entry	Buffer ^a^	Additives ^b^	pH	Presence of Soluble Protein After Lysis ^c^	IMAC Compatible ^d^
1	Tris	NaCl, L-Arg, L-Glu	8.5	×	n.d.
2	EPPS	NaCl, L-Arg, L-Glu	8.5	×	n.d.
3	BisTris	NaCl, L-Arg, L-Glu	8.5	×	n.d.
4	CHES	NaCl, L-Arg, L-Glu	9.5	√	×
5	Tris	NaCl, CHAPS (5 mM), Triton X-100 (1%)	8.5	√	×
6	CHES	NaCl	9.5	√	×
7 ^e^	Tris	NaCl	7.5	×	n.d.
8	Tris	NaCl	8.5	√	√
9	AMP	NaCl	8.5	√	×
10	AMP	NaCl	9.0	√	×
11	AMP	NaCl	9.5	√	×
12	AMP	NaCl	10.0	√	×
13	AMP	NaCl	10.5	√	×

^a^ All buffers were used at a 50 mM concentration unless otherwise stated. Tris: tris(hydroxymethyl)aminomethane; EPPS: 4-(2-hydroxyethyl)-1-piperazinepropanesulfonic acid; BisTris: 2,2-bis(hydroxymethyl)-2,2′,2′′-nitrilotriethanol; CHES: N-cyclohexyl-2-aminoethanesulfonic acid; AMP: 2-amino-2-methyl-1-propanol. ^b^ Additive concentrations used: NaCl, 100 mM; L-Arg, 500 mM; L-Glu, 500 mM. ^c^ As determined by the presence of a distinct band on the SDS-PAGE analysis at the expected molecular weight in the lane(s) corresponding to the soluble fraction of the cell lysate. ^d^ Compatibility was assessed based on whether fractions eluted after IMAC showed a single protein band of the correct molecular weight by SDS-PAGE analysis. n.d.: not determined, as these buffers did not give soluble silicatein. ^e^ Buffer conditions from ref. [15].

**Table 2 biomolecules-10-01209-t002:** Buffer compositions investigated during cell lysis and SAC purification of TF-Silα-Strep.

Entry	Buffer ^a^	Additives ^a^	pH	Presence of Soluble Protein After Lysis ^b^	SAC Compatible ^c^
1	CHES	NaCl, L-Arg, L-Glu	9.5	√	√
2	Tris	NaCl, CHAPS (5 mM), Triton X-100 (1%)	8.5	√	√
3	CHES	NaCl	9.5	√	√
4	Tris	NaCl	8.5	√	√
5	AMP	NaCl	8.5	√	√
6	AMP	NaCl	9.0	√	√
7	AMP	NaCl	9.5	√	√
8	AMP	NaCl	10.0	√	√
9	AMP	NaCl	10.5	√	√

^a^ All concentrations and abbreviations are the same as those listed in Table 1. ^b^ As determined by the presence of a distinct band on the SDS-PAGE analysis at the expected molecular weight in the lane(s) corresponding to the soluble fraction of the cell lysate. ^c^ Compatibility was assessed based on whether fractions eluted after SAC showed a single protein band of the correct molecular weight by SDS-PAGE analysis.

**Table 3 biomolecules-10-01209-t003:** Buffer compositions used for biophysical measurements.

Buffer	Additives	pH
Potassium phosphate (100 mM)	KCl (20 mM)	7.0
Tris-HCl (50 mM)	NaCl (100 mM)	8.5
AMP (50 mM)	NaCl (100 mM)	9.0

**Table 4 biomolecules-10-01209-t004:** Comparative summary of protein production yields for the Silα fusion proteins.

Protein	Buffer	pH	Isolated Yield (nmol L^−1^ of Cell Culture) ^a^
His_6_-Silα	Tris (50 mM), NaCl (100 mM)	8.5	8
His_6_-TF- Silα	Potassium phosphate (100 mM)	8.0	6
	Tris (50 mM), NaCl (100 mM)	8.5	177
TF-Silα-Strep	Tris (50 mM), NaCl (100 mM)	8.5	777
	AMP (50 mM), NaCl (100 mM)	9.0	1540

^a^ Calculated based on the molecular weights of each protein: His_6_-Silα, 25.2 kDa; His_6_-TF-Silα, 79.1 kDa; TF-Silα-Strep, 74.6 kDa.

**Table 5 biomolecules-10-01209-t005:** Polydispersity and polydispersity index (PDI) of each protein, as predicted from the DLS data. Data shown are the average ± the standard deviation of two protein samples.

pH	His_6_-TF-Silα		TF-Silα-Strep	
	% Polydispersity	PDI	% Polydispersity	PDI
7.0	93.8 ± 27.3	1.65 ± 0.57	41.6 ± 6.4	0.326 ± 0.047
8.5	130.3 ± 15.6	2.29 ± 0.07	49.3 ± 1.7	0.302 ± 0.062
9.0	61.7 ± 21.5	1.47 ± 0.68	34.5 ± 6.7	0.152 ± 0.079

**Table 6 biomolecules-10-01209-t006:** Michaelis-Menten constants for the hydrolysis of TBDMS-ONp catalysed by His_6_-TF-Silα and TF-Silα-Strep. The errors represent standard deviations.

Enzyme	*K_M_* (µM)	*k_cat_* (min^−1^)	*k_cat_/K_M_* (min^−1^ µM^−1^)
His_6_-TF-Silα	22.4 ± 2.2 ^a^	988 ± 416 ^a^	44.1
TF-Silα-Strep	88.2 ± 34.6	0.150 ± 0.033	0.00170

^a^ Data taken from ref. [13].

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
