# Peer review of "Improved Production and Biophysical Analysis of Recombinant Silicatein-α"

_biomolecules, 2020, doi:10.3390/biom10091209_

Round 1
Reviewer 1 Report
The manuscript by Sparkes and colleagues concerns the over-expression of recombinant silicatein. They successfully identify constructs and conditions leading to very high yields of more stable protein but unfortunately catalytic activity is almost abolished for these constructs (decreased 7000-fold).
The paper is well-written and clear. Appropriate context and background are provided in the introduction, with a suitable survey of the relevant literature. Methods are provided at sufficient detail for others to understand the work and repeat it. The experiments are carried out carefully, and appropriate conclusions are drawn. It is a timely study that certainly makes a nice contribution to the field and will allow others to work with this protein too.
- Three constructs were tested, and the results are clear: adding the C-terminal Strep tag improves things dramatically. However, a direct comparison to a C-terminal His-tag, or indeed N-terminal strep-tag, is lacking. Without this nothing can be said about the problems the His-tag may cause; it might be that when a His-tag is added to the C-terminus this brings about the same improvements. The authors say on P6 (and elsewhere) that “the His6 tag negatively affects solubility” but this conclusion cannot be supported without trialling at least (i) a C-terminal His-tag, (ii) Protein purified by an N-terminal His tag that has subsequently been cleaved and removed, and (iii) an N-terminal strep tag. This does not undercut the main results, but I do think some of the interpretation is a bit strong. Please amend the text accordingly.
- The SDS-PAGE gels shown in Fig. S2 are a key component of this work, and it is a surprise to see them tucked away in the supplement. It would be good if the authors could at least move the post-column gels into the main body of the paper (and re-label Fig. S2(b) since this is strep-tag, not IMAC). It would be even better to see an absolute ‘killer’ gel showing a great thick band of highly pure strep-tagged protein.
- One intriguing aspect is that the authors isolate 90 mg of strep-tagged protein in a single prep from 800 ml media. However according to the manufacturer the binding capacity of a 5ml StrepTrap HP column is only about 30 mg. Can the authors just comment to resolve this please?
- The CD data are impossible, please recalculate. The absolute lowest MRE value one can achieve for a protein is about -38,000. The values given here are about 10x too large. This is essential and a condition of publication.
- Many readers will have expected thermal unfolding studies to use CD and particle dispersity to be determined by gel filtration, as is traditional for purified proteins. The authors could comment on why they opted not to use these techniques.
Reviewer 2 Report
The manuscript ‘Improved production and biophysical analysis of recombinant silicatein-α’ presented a detailed overview of efforts towards streamlining the expression and purification of this unique silicon-oxygen bond-forming enzyme. The work related to improvements in protein expression and purification was presented very technically and concisely. It was a bit disappointing to read that the most stable and highest yielding protein construct (TF-Silα-Strep) failed to retain strong catalytic activity in the assays examined, however this should not disqualify this manuscript from further consideration in this journal. The trade off between protein solubility and activity is a difficult balance, and this work lays the foundation to help establish an ideal construct of this industrially useful enzyme.
Despite this interesting work, there were a few major concerns that should be addressed prior to publication. The most significant ones in my opinion relate to the claims of the N-terminal His-tag being hugely significant for enzymatic catalysis, and the depiction of the data presented in the supporting information. Following revision of these comments, I would be happy to further consider this manuscript for publication in Biomolecules.
Major comments:
One of the major points made in this manuscript is the importance of the protein affinity tag for both solubility and activity. While this was strongly supported in the protein expression/solubility/aggregation sections of the manuscript, the claims made in the activity assay sections were broad and need additional experimental support. Attributing a large amount of the catalytic activity of this enzyme construct to the N-terminal His tag may be true, however there were not enough experiments shown to firmly convince this reviewer that this is the case. The hypothesis that the His-tag may be promoting catalysis has literature precedence; however the degree of difference established here suggests more than just His-tag catalytic promotion is involved in the 4+ order of magnitude fold difference in kinetic parameters between the His-TF-Silα and TF-Silα-Strep constructs. Under the current conditions this implies that the His-tag is facilitating the majority of the catalytic work and diminishes the role of the known enzyme active site itself. Possible additional explanations for this discrepancy in activity may be due to the introduction of inhibitory effects from the C-terminal Strep tag or that a free C-terminus is required for efficient catalysis. It would be incredibly beneficial to have a parallel enzyme assay with the Strep tag cleaved off from the TF-Silα-Strep following Strep affinity purification to eliminate this as a variable. A complimentary approach would involve cleaving the N-terminal His-tag off of the His-TF-Silα in order to support the hypothesis of its significance in catalysis. Alternative potential constructs to explore the importance of affinity tags on activity would be an N-terminal Strep-TF-Silα construct, or a His-TF-Silα-Strep fusion. If these experiments cannot be completed or wish to be reserved for follow up studies, the claims made in this manuscript that the His-tag is solely responsible for this dramatic rate acceleration need to be significantly walked back as other explanations remain unexplored.
Figures S1 and S2 – The gel images are depicted in a very haphazard manner, particularly in the top row of S1 and in the TF-Silα-Strep construct of S2. The marker ladder appears to be cut and pasted next to other gel lanes that were run on different gels, creating a bit of distrust in the rigor presented within this section of the manuscript. Also is there a reason that conditions for entry 7 (the one successful entry from Table 1) are not depicted in Figure S1? This seems like a significant omission to not present the one successful case in this format. Overall, I would highly recommend depicting these gel images in a more homogenous format (similar to conditions 9-11 shown on the bottom row of Fig. S1) in order to improve transparency and reader confidence in the results. Lastly, using a different labeling format (instead of numbers) or colors for the individual lanes would help minimize confusion between the entry numbers listed at the bottom of the individual gel images.
Figure S3 and Table 5 kinetic data – The Michaelis-Menten plot does not appear to have error bars and the substrate concentrations tested do not appear to saturate the enzyme. The previous study (ref 13) reported that the His-TF-Silα required at least 200 μM to reach enzyme saturation, while the provided plot stops at half that concentration. Also the numbers of technical replicates performed are missing. These factors should be addressed to most accurately compare the activity of these protein constructs.
Minor comments:
Tables 1 and 2 – Were other commonly used techniques like addition of glycerol or low concentrations of reducing agents (DTT, TCEP) to buffers used to help improve solubility and stability? These would have been the first things I would have attempted and it might be useful to mention why they were not chosen.
Line 213 – L-glutamine is written here whereas all other additive descriptions (line 223 or Table 1) indicated that L-Glu (glutamic acid) was added.
Line 222/223 and Table 1 – The information provided in lines 222-223 should be included in Table 1 for comparison purposes and cited accordingly. Even though it was not conducted within this current study, it helps to show additional conditions that have been attempted to purify this difficult protein construct and would improve transparency.
Line 253 – grammatical error - ‘no visible aggregates were ever observed’
Line 260/261 – The stock formatting description for Tables is left at the end of the Table 2 caption.
Figure 4c – It would be useful to have a description in the figure legend as to why the pH 7.0 dots become dashed after 60 °C.
Reviewer 3 Report
Please see the attached file.

Round 2
Reviewer 2 Report
The authors have improved their manuscript to increase transparency and further explain and support their claims. All of the major concerns raised upon first review have been addressed in some capacity. While it would be great to know the activity results of the various other Silα affinity tag constructs, given the current limitations due to COVID-19, I believe that the adjustments to the manuscript are sufficient to warrant publication in Biomolecules.
Author Response
We thank the reviewer for their review. They have not raised any comments that require a specific response, and have recommended the manuscript is suitable for publication.
Reviewer 3 Report
The authors made almost appropriate corrections according to the reviewer' comments. Regarding the second point of the major issues, the reviewer stated the need for appropriate negative controls in the measurement of hydrolysis activity. This means that the reviewer asked the authors to represent the negative control data. The authors responded to this issue by citation, but did not provide data for negative controls. In the authors’ response to reviewer 2, the authors claim that the additional experiment is difficult due to the current situation against COVID19, and the reviewer understands it. Therefore, the reviewer thinks that the amendment is within the scope of acceptance. The final decision is left to the editor.
I would like to point out some minor issues as follows:
L36 Species name “Suberites domuncula” should be italicized. Genus name should be abbreviated after the second time. (L242 and somewhere else)
L91 “IMAC” has already defined at L68.
Manuscripts should be thoroughly checked to be written according to the guideline of the journal.
